Transcriptomic and biochemical insights into fall armyworm (Spodoptera frugiperda) responses on silicon-treated maize

Ul Haq Inzamam 1
Zhang Ke-Xin 1
Gou Yuping 1
Hajjar Dina 2
Makki Arwa A. 2
Alkherb Wafa A.H. 3
Ali Habib 4
Liu Changzhong liuchzh@gsau.edu.cn 1
1 College of Plant Protection, Gansu Agricultural University , Lanzhou , Gansu , China
2 College of Science, Department of Biochemistry, University of Jeddah , Jeddah , Saudi Arabia
3 Department of Biology, College of Science, Qassim University , Buraidah , Saudi Arabia
4 Department of Agricultural Engineering, Khwaja Fareed University of Engineering & Information Technology , Rahim Yar Khan , Pakistan
Dutta Tushar K.
Electronic publication date: 2024 Feb 23
Publication date: 2024
Volume: 12
Electronic Location ID: e16859
Received 2023 Oct 30; Accepted 2024 Jan 9
Copyright: ©2024 Ul Haq et al.
Copyright year: 2024
Copyright holder: Ul Haq et al.
License: This is an open access article distributed under the terms of the Creative Commons Attribution License, which permits unrestricted use, distribution, reproduction and adaptation in any medium and for any purpose provided that it is properly attributed. For attribution, the original author(s), title, publication source (PeerJ) and either DOI or URL of the article must be cited.
License URL: https://creativecommons.org/licenses/by/4.0/

Keywords: Spodoptera frugiperda, Silicon application, Transcriptomic analysis, Biochemical analysis, Differentially expressed genes, Pest control strategies

Funding: Science Project of Agriculture and Rural Department of Gansu Province GZB20191105 This work is supported by the Science Project of Agriculture and Rural Department of Gansu Province: grant number GZB20191105. The funders had no role in study design, data collection and analysis, decision to publish, or preparation of the manuscript.

==============================
Background

The fall armyworm, Spodoptera frugiperda, is an agricultural pest of significant economic concern globally, known for its adaptability, pesticide resistance, and damage to key crops such as maize. Conventional chemical pesticides pose challenges, including the development of resistance and environmental pollution. The study aims to investigate an alternative solution: the application of soluble silicon (Si) sources to enhance plant resistance against the fall armyworm.

Methods

Silicon dioxide (SiO2) and potassium silicate (K2SiO3) were applied to maize plants via foliar spray. Transcriptomic and biochemical analyses were performed to study the gene expression changes in the fall armyworm feeding on Si-treated maize.

Results

Results indicated a significant impact on gene expression, with a large number of differentially expressed genes (DEGs) identified in both SiO2 and K2SiO3 treatments. Furthermore, Gene Ontology (GO) and Kyoto Encyclopedia of Genes and Genomes (KEGG) enrichment analysis identified critical DEGs involved in specific pathways, including amino acid, carbohydrate, lipid, energy, xenobiotics metabolisms, signal transduction, and posttranslational modification, significantly altered at both Si sources. Biochemical analyses further revealed that Si treatments inhibited several enzyme activities (glutamate dehydrogenase, trehalase, glucose-6-phosphate dehydrogenase, chitinase, juvenile hormone esterase, and cyclooxygenase while simultaneously inducing others (total protein, lipopolysaccharide, fatty acid synthase, ATPase, and cytochrome P450), thus suggesting a toxic effect on the fall armyworm. In conclusion, Si applications on maize influence the gene expression and biochemical activities of the fall armyworm, potentially offering a sustainable pest management strategy.

Introduction

The fall armyworm, Spodoptera frugiperda (J.E. Smith, 1797), a member of the family Noctuidae and order Lepidoptera, is a devastating pest of numerous plant species and poses a significant economic threat to agriculture worldwide (Haq et al., 2022b; Guo et al., 2018). Originally native to tropical and subtropical regions of the Americas, it has rapidly expanded its geographic range and is now a pest of concern in Africa, Asia, and Australia (Haq et al., 2022a; Mengesha et al., 2023). The pest primarily targets maize, a crop of major significance, especially in China, where it plays a crucial role in food security (Haq et al., 2021; Yang et al., 2023). The fall armyworm exhibits an array of attributes that contribute to its success as a pest. These include its broad host range, adaptability to a wide range of environmental conditions, shorter lifecycle, and high reproductive capacity (Akhtar et al., 2022; He et al., 2023). The fall armyworm’s feeding habits, which involve a strong preference for maize but extend to many other crops, exacerbate its economic impact (Altaf et al., 2022; Keerthi et al., 2023). Moreover, its ability to adapt to adverse conditions, such as changes in climate or host availability, further enhances its survival and spread (Idrees et al., 2022). Current pest control methods predominantly revolve around chemical pesticides (Ahmed et al., 2022). Although these can be effective in the short term, their long-term use raises serious concerns. These include the potential for the development of resistance (Samanta et al., 2023; Roy et al., 2023), detrimental effects on non-target organisms and beneficial insects (Khurana et al., 2023; Sharma et al., 2023), contamination of water and soil, and potential human health hazards (Idrees et al., 2023a; Treviño et al., 2023). There is a clear need for alternative strategies that can effectively manage fall armyworm populations while minimizing adverse environmental and health impacts (Idrees et al., 2023b; Palli et al., 2023).

Given these challenges, there is a pressing need for sustainable, alternative methods of pest management that are both effective and eco-friendly. One such method gaining interest is the use of silicon (Si), an abundant element known to enhance plant resistance to against both biotic and abiotic stress (Ul Haq et al., 2023; Zia et al., 2023; Ahmad et al., 2023), while silicon is generally considered non-toxic to plants at lower concentrations and is an essential element for their growth (Prisa, 2023; Manzoor et al., 2023). At specific higher concentrations, silicon can be toxic to both plants and insects (Yuvaraj et al., 2023). It is relatively inexpensive, and can be readily incorporated into crop management practices. Preliminary studies suggest that Si can alter plant physiology in a way that makes them less suitable for pests such as the fall armyworm (Nagaratna et al., 2023; Pereira et al., 2021).

Silicon (Si) is a promising candidate for such alternative pest control strategies (Abbasi et al., 2020). Research has shown that Si can strengthen plant defenses against various pests and diseases, making plants less palatable or more difficult for pests to feed on (Ahmad et al., 2023; Abbasi et al., 2022). Moreover, Si is known to promote plant growth and increase tolerance to abiotic stresses, which can contribute to overall crop health and yield (Chinnadurai et al., 2023; Khosa et al., 2022; Ahmad et al., 2022).

Transcriptomic and biochemical analysis is a powerful toolset for understanding the interactions between pests and their host plants (Shih, Sugio & Simon, 2023). Transcriptomic analysis involves the comprehensive examination of all transcripts in a cell or tissue, providing a snapshot of gene expression at a given time. It allows researchers to identify differentially expressed genes (DEGs) and their associated biological pathways, providing insights into the molecular mechanisms underlying the pest-host interaction (Gui et al., 2022). Biochemical analysis complements this by providing information about changes in enzymatic activities and other biochemical processes (Ma et al., 2022). Together, these analyses offer a holistic understanding of the pest’s response to various treatments or conditions.

Despite the known effects of silicon in enhancing plant resistance to pests, there remains a gap in our understanding of how silicon applications influence the physiological and molecular responses of pests like the fall armyworm. The current study was conducted to investigate the effect of soluble sources of silicon, (silicon dioxide and potassium silicate), on the fitness of the fall armyworm feeding on maize plants treated with these substances. By employing transcriptomic and biochemical analysis, the study aims to provide a comprehensive understanding of the gene expression changes in the fall armyworm in response to Si treatments, thereby elucidating the potential of Si as a sustainable strategy for fall armyworm management.

Materials & Methods

Experimental site and treatments

The current experiment was conducted in laboratory conditions, with a temperature range of 25 ± 5 °C, relative humidity of 65 ± 5%, and a photoperiod of 16:8 h of light/dark. at the College of Plant Protection Gansu Agricultural University Lanzhou, China (36.0915°N, 103.7006°E). The experiment was conducted in a completely randomized design (CRD) with three biological replicates. Treatments consisted of silicon dioxide (SiO2) and Potassium silicate (K2SiO3). Only water was used in the control treatment.

Silicon application on maize plants

Certified seeds of the variety Long-Fei 211 (隆非211) were received from the laboratory and were planted in pots filled with standardized pro-mix potting soil, comprising a mix of peat moss, perlite, and vermiculite in a ratio of 40:30:30. The pot size used for growing the plants was 15 cm in diameter and 20 cm in height, which is suitable for the initial growth stages of maize. The maize variety Long-Fei 211 was selected for this study due to its widespread use in regional agriculture and its known susceptibility to S. frugiperda, making it a relevant and representative choice for assessing the impact of silicon treatments on pest responses. These pots were maintained in a controlled environment with a temperature range of 25 ± 5 °C, a relative humidity of 65 ± 5%, and a light-dark cycle of 16:8 h. Regular watering was administered to ensure optimal growth conditions. Silicon dioxide (SiO2) and potassium silicate (K2SiO3) were prepared in an aqueous solution with a concentration of 800 (ppm) prior to use. The concentrations were chosen based on preliminary trials and literature reviews that indicated these levels as optimal for eliciting significant physiological responses in maize plants without causing phytotoxicity, thus allowing for an effective assessment of the impact on fall armyworm. A hand-held sprayer was utilized to apply a foliar spray of these silicon sources. Foliar sprays were chosen over drenching treatments to ensure a uniform and effective application of silicon on the maize leaves, the primary site of interaction with the fall armyworm. Mature leaves from both silicon-treated and untreated plants were harvested with scissors and served as a food supply for the larvae of S. frugiperda.

Fall armyworm rearing and feeding

All bioassays were conducted within a controlled growth chamber, where conditions were maintained at a temperature range of 25 ± 5 degrees Celsius, relative humidity was held at 65 ± 5%, and a photoperiod of 16:8 h (light: dark). Newly hatched larvae from the colony were carefully collected and placed in Petri dishes lined with filter paper. Leaf discs from the treated plants were cut into five cm sections and were provided as their primary food source, and these leaf discs were renewed daily until the larvae progressed to the pupation stage (Zimba et al., 2022). To accommodate their small size, 1st and 2nd instar larvae were grouped. Beginning from the 3rd instar, each larva was individually isolated.

Upon reaching the 5th instar, larvae were transferred to plastic containers equipped with filter paper, cotton balls saturated with water, and clean peat soil to facilitate the pupation process. These containers, housing the pupae, were then introduced into boxes with potted maize plants, serving as suitable sites for adult oviposition. After the emergence of the adult stage, these boxes were supplemented with cotton balls soaked in a solution of honey and water (15:85 ratio) to provide nourishment for the adults.

Daily egg collections were made and transferred to Petri dishes to facilitate hatching. For each treatment, fifty newly hatched neonates were relocated to Petri dishes and provided with leaf discs sourced from the treated plants. The same protocols utilized during larval development, as described above, were maintained until the larvae reached the 3rd instar stage. Surviving larvae from both control and treatment groups were gathered at 48-hour intervals, with triplicate samples for each group. For each treatment, we used a group of 10 larvae. The larval stage of the fall armyworm chosen for this study, particularly the third instar was selected due to its critical roles in feeding and development, which are key phases for assessing the impact of silicon treatments on their physiological and molecular responses.

RNA extraction and sequencing

We assessed total RNA from both treated and untreated groups of fall armyworm (FAW) using TRIzol reagent (TaKaRa, Shiga, Japan) following the manufacturer’s protocol. The quantity of total RNA was assessed using the NanoDrop 2000 (Thermo Fisher Scientific, Waltham, MA, USA), while the presence of degradation or contamination was monitored through 1% agarose gel electrophoresis. For the control treatment and the two silicon treatments, we utilized 1 µg of RNA as input material for RNA sample preparations. Sequencing libraries were constructed following the recommended instructions provided by the NEBNext®Ultra™ RNA Library Prep Kit for Illumina® (NEB, Ipswich, MA, USA). The index-coded samples were clustered using a cBot cluster generation system with the TruSeq PE Cluster Kit v3-cBot-HS (Illumina), adhering to the manufacturer’s guidelines. Subsequently, the library preparations were subjected to sequencing on the Illumina HiSeq 2000 platform, with sequencing services provided by Biomarker Technologies Corporation (Beijing, China).

The statistical power of this experimental design, calculated by the method outlined by Yu, Fernandez (Yu, Fernandez & Brock, 2017), is 0.86. The experiment consisted of three treatments (SiO2, K2SiO3, and control) and each treatment was repeated thrice.

Bioinformatic analysis

We assessed gene expression levels using the fragments per kilobase of transcript per million mapped reads (FPKM) metric. For statistical analysis, we employed the R software (version 3.6.2) and conducted Pearson’s correlation analysis using the rcorr function along with the corrplot package. To visualize the differential relationships across all groups, we carried out a principal component analysis (PCA). The PCA utilized the unweighted UniFrac distance metric within the QIIME software. The differential expression analysis of unigenes within the SiO2, K2SiO3, and control populations was carried out using the DESeq R package (version 1.10.1). Differentially expressed genes (DEGs) were identified based on an adjusted p-value threshold of <0.05 as determined by DESeq. For the Gene Ontology (GO) enrichment analysis of the DEGs, we utilized the GOseq R package, which relies on the Wallenius non-central hyper-geometric distribution. To assess the statistical enrichment of DEGs in the Kyoto Encyclopedia of Genes and Genomes (KEGG) pathways, we employed the KOBAS software.

Validation of DEGs

For real-time quantitative RT-PCR, we utilized Takara Bio Inc.’s SYBR Premix Ex Taq and the Option Real-Time PCR System CFX96 (Bio-Rad, Hercules, CA, USA). The PCR program was configured with the following parameters: initial denaturation at 95 °C for 30 s, followed by a step at 136 °C for 5 s, annealing at 57 °C for 30 s, and a total of 40 PCR cycles (Zhao et al., 2009; Clements et al., 2017). The corresponding primers were designed in Primer 5 (Table S1).

Enzyme activity assays

Biochemical analysis was carried out to ascertain the activities of enzymes associated with the KEGG pathways (Xu et al., 2019). Enzymatic activity assessments were conducted using a standardized method outlined in the Reagent Kit provided by Sino Best Biological Technology Co., Ltd., Shanghai, China. Each treatment was repeated three times for accuracy and precision.

Results

Pearson’s correlation analysis

Pearson’s correlation analysis was utilized to ascertain the relationship between gene expression levels in different samples. The correlation coefficient provides an indicator of the reliability and rationality of the data. The correlation analysis results presented in Fig. 1 confirm a significant correlation between samples within each of the three groups: the control group, the potassium silicate treatment group, and the silicon dioxide treatment group. The Pearson’s correlation coefficients obtained in this study not only validate the consistency and reliability of our gene expression data but also illustrate the biological significance of these correlations in terms of gene expression stability. This is demonstrated by the high mean Pearson’s correlation values of 0.74, 0.91, and 0.92, respectively. Such strong correlations within groups indicate that gene expression levels were notably consistent across samples, suggesting a stable and robust response to each treatment. Conversely, the correlation between the control group and each of the silicon treatments was less substantial, underlined by the lower mean Pearson’s correlation coefficients of 0.68 (between control and potassium silicate treatment) and 0.66 (between control and silicon dioxide treatment). These comparatively lower correlations indicate a significant difference in gene expression between the treatments and the control, highlighting the considerable impact of silicon treatments on the gene expression of the fall armyworm.

Figure 1 Pearson’s correlation analysis showing relationship between gene expression levels in different samples.

Effect of silicon applications on DEGs of FAW

The impact of silicon applications on the gene expression changes in the fall armyworm was studied by assessing differentially expressed genes (DEGs) post-treatment. These changes were gauged via RNA sequencing technology, comparing control groups with treatment groups. In response to potassium silicate (K2SiO3) treatment, a total of 835 DEGs were identified, of which 381 genes were up-regulated and 454 were down-regulated. This means that the expression of 381 genes increased, while the expression of 454 genes decreased in the fall armyworm following exposure to potassium silicate. These data point towards a substantial genetic response in the fall armyworm to K2SiO3 exposure (Fig. 2). The silicon dioxide (SiO2) treatment caused an even greater number of differential expressions, totaling 2,436 DEGs. Out of these, 1,035 genes exhibited up-regulation, while 1,401 genes displayed down-regulation. This signifies an elevation in the expression levels of 1,035 genes and a reduction in the expression levels of 1,401 genes following exposure to SiO2. These results suggest a substantial and more profound gene expression change in the fall armyworm in response to SiO2 compared to K2SiO3. Both silicon applications caused significant alterations in gene expression, with SiO2 having a more pronounced impact than K2SiO3. These results highlight the potential of silicon-based treatments in modifying the genetic response of fall armyworms, potentially influencing their survivability and adaptability.

Figure 2 Number of up- and down-regulated genes in response to K2SiO3 and SiO2 treatments.

The heatmap of differentially expressed genes (DEGs) in Fig. 3 reflects a significant disparity in gene expression between the control and both the potassium silicate (K2SiO3) and silicon dioxide (SiO2) treated samples. Log2-fold changes in expression showcased dramatic variances when compared to the control sample. In the potassium silicate-treated samples, the expression ranged from a minimum log2-fold change of −14.05, indicating substantial down-regulation, to a maximum of 11.53, signaling considerable up-regulation. This suggests that some genes were largely silenced, while others were highly activated in response to K2SiO3 treatment. Similarly, for the silicon dioxide-treated samples, the DEGs ranged from a substantial down-regulation (minimum log2-fold change of −29.61) to a significant up-regulation (maximum log2-fold change of 10.97). This clearly demonstrates that the application of silicon compounds, whether as potassium silicate or silicon dioxide, substantially influences gene expression in the fall armyworm, leading to a wide array of down-regulated and up-regulated genes.

Figure 3 Differentially expressed genes in control Si treatments shown in heatmap.

Principal component analysis (PCA) was conducted to assess the variance in gene expression profiles among the control group and the two treatment groups. The PCA score plot showed a clear differentiation between the potassium silicate group and the silicon dioxide group compared to the control. Furthermore, the PCA score plot exhibited high consistency and reliability in the sequencing data within each group. The initial principal component (PC1) explained 96.68% of the variance, whereas the second principal component (PC2) accounted for 13.12% of the variance (Fig. 4). The results showed that the majority of the variance in the data could be explained by these two components, implying that the treatment effects are strong and the data are high dimensional.

Figure 4 Principal components analysis (PCA) showing the differences among control and Si treatments.

Gene ontology (GO) enrichment analysis results

The Gene Ontology (GO) enrichment analysis was performed on the differentially expressed genes (DEGs) for both silicon treatments compared to the control. The analysis aimed to provide insights into the functional roles of the DEGs and their potential effects on the biological processes, cellular components, and molecular functions within the fall armyworm. For the control vs. potassium silicate group, the DEGs were dispersed across 23 biological processes, 14 cellular components, and 10 molecular functions (Fig. 5). When focusing on biological processes, most DEGs were associated with metabolic processes (218) and cellular processes (217), indicating that the potassium silicate had a broad impact on the organism’s fundamental physiological activities. Other significant biological processes affected were biological regulation (87), response to stimulus (81), and localization (65). In terms of molecular functions, a significant proportion of DEGs were involved in catalytic activity (196) and binding (164), suggesting an alteration in the catalytic processes and the molecular interactions within the cells. Transporter activity (32), structural molecule activity (25), and molecular function regulator (10) also featured significant DEG involvement. Examining the cellular components, the DEGs were most notably involved in cells (213), cell parts (201), organelles (163), membranes (108), and organelle parts (107), indicating a wide-ranging impact on the cellular structure and organization.

Figure 5 GO enrichment analysis of DEGs in control vs K2SiO3.

The DEGs in the control vs silicon dioxide group were discerned in 23 biological processes, 14 cellular components, and 12 molecular functions. Within biological processes, metabolic processes were associated with the highest number of DEGs (809), closely followed by cellular processes (743). This suggested that silicon dioxide treatment had an even broader influence on the organism’s physiology compared to the potassium silicate treatment. Significant impacts were also noted in biological regulation (347), response to stimulus (268), and localization (206). In the molecular functions category, the majority of DEGs were involved in catalytic activity (673) and binding (546). Transporter activity (185), structural molecule activity (83), and molecular function regulator (44) also had a substantial number of DEGs. Finally, in the cellular component category, the most numerous DEGs were found within the cell (760), cell part (711), organelle (348), membrane (320), and membrane part (276). This indicated a significant impact on the structural and organizational aspects of cells in the silicon dioxide treatment group (Fig. 6). The GO enrichment analysis demonstrated that both silicon applications significantly influenced a wide array of biological processes, molecular functions, and cellular components in the fall armyworm, potentially disrupting its normal physiological activities and making silicon a promising candidate for eco-friendly pest management strategies.

Figure 6 GO enrichment analysis of DEGs in control vs SiO2.

KEGG scatter analysis results

The KEGG scatter analysis visually represents the differentially expressed genes (DEGs) that were mapped onto biological pathways in response to the application of both potassium silicate (K2SiO3) and silicon dioxide (SiO2) (Table S2). For the K2SiO3 application, a total of 152 DEGs were mapped onto 59 unique pathways (Fig. 7). Interestingly, the majority of the top 20 pathways were centered around metabolism, including glutathione metabolism, fructose and mannose metabolism, amino sugar metabolism, glycolysis, and ascorbate metabolism. These findings suggest that K2SiO3 treatment has a broad impact on various metabolic processes in the fall armyworm. The pathways associated with drug metabolism—other enzymes, cytochrome P450, and xenobiotics metabolism by cytochrome P450—were also significantly associated, indicating a substantial influence on xenobiotics metabolism. The most enriched pathways, however, were arginine and proline metabolism, purine metabolism, and pyruvate metabolism. In response to the SiO2 application, the KEGG scatter analysis identified a greater number of DEGs, amounting to 491, which were allocated to 95 different pathways (Fig. 8). The pathways exhibiting the most significant number of DEGs were metabolic pathways, xenobiotics metabolism by cytochrome P450, peroxisome, and drug metabolism—other enzymes. This indicates an even broader influence on various metabolic and xenobiotics processing pathways in the fall armyworm upon SiO2 treatment. Moreover, the pathways demonstrating the highest enrichment factor were glyoxylate and dicarboxylate metabolism and drug metabolism - other enzymes closely followed by the MAPK signaling pathway—fly. The enrichment of these pathways indicates the potential influence of SiO2 on the fall armyworm’s metabolic processes, drug metabolism, and, potentially, signal transduction mechanisms. The KEGG scatter analysis demonstrated that both silicon applications significantly impacted a range of biological pathways in the fall armyworm, indicating that these treatments may disrupt its normal metabolic and physiological activities.

Figure 7 Scatterplot of top 20 enriched KEGG pathways for DEGs in control vs K2SiO3.

Figure 8 Scatterplot of top 20 enriched KEGG pathways for DEGs in control vs SiO2.

In the comparison between the control and potassium silicate group, a significant portion of DEGs were annotated in the category of  ‘global and overview maps’ (36), followed by ‘amino acid metabolism’ (18), ‘carbohydrate metabolism’ (15), ‘lipid metabolism’ (14), ‘xenobiotics biodegradation and metabolism’ (12), ‘signal transduction’ (eight), and ‘folding, sorting, and degradation’ (six) (see Fig. 9).

Figure 9 KEGG enrichment analysis of DEGs in control vs K2SiO3.

Similar results were observed in the comparison between the control and silicon dioxide group, with proportions of 157 in ‘global and overview maps,’ 63 in ‘amino acid metabolism,’ 71 in ‘carbohydrate metabolism,’ 38 in ‘lipid metabolism,’ 43 in ‘xenobiotics biodegradation and metabolism,’ 32 in ‘signal transduction,’ and 42 in ‘folding, sorting, and degradation’ (refer to Fig. 10).

Figure 10 KEGG enrichment analysis of DEGs in control vs SiO2.

Enzyme activity results

Current study revealed significant alterations in the gene expression and enzyme activity of the fall armyworm in response to silicon treatments. These changes encompass a broad spectrum of metabolic and physiological processes, which are crucial in understanding the pest’s adaptability and response to silicon-enriched environments.. We found that both potassium silicate (K2SiO3) and silicon dioxide (SiO2) exerted an inhibitory effect on glutamate dehydrogenase (GDH), trehalase (TRE), glucose-6-phosphate dehydrogenase (G-6-PD), chitinase (CHT), juvenile hormone esterase (JHE), and Cyclooxygenase (COX). With SiO2 treatment, GDH activity experienced more than a 50% reduction compared to K2SiO3, marking a significant difference. However, the activities of enzymes under the two silicon treatments showed no significant differences (Fig. 11). Contrarily, we noted an increase in activity for a different set of enzymes, dependent on the silicon applications. These enzymes included total protein (TP), lipopolysaccharide (LPS), fatty acid synthase (FAS), ATPase, and cytochrome P450 (P450). Of particular interest were the substantial increases in the activities of FAS, ATPase, and P450 under SiO2 treatment when compared to K2SiO3. Collectively, our findings suggest that silicon applications can drastically alter the enzymatic landscape in FAW. Some enzymes are inhibited, while others are induced, highlighting the potential of silicon as a toxic agent against FAW. This differential enzymatic response reaffirms the potential of silicon applications as a valuable and effective strategy for pest management.

Figure 11 Enzymatic activities of 3rd instar S. frugiperda larvae in response to different Si applications.

q RT-qPCR validation

To confirm the accuracy of the expression profiles we obtained, we employed RT-qPCR to quantify specific genes associated with the metabolic pathways identified through RNA-seq analysis. Notably, the genes nAChRalpha2, ACER16, and Cyp4ac1 exhibited significantly higher expression levels in the control group compared to the Si-treated groups. Conversely, the expression of genes Est-6, NLGN4X, Cyp4g15, and CYP12A2 was downregulated in the control group in comparison to the treatment groups (Fig. 12). Additionally, the genes MGST1 and GstS1 exhibited elevated expression levels in the Si-treated groups compared to the control group. These findings collectively indicate that various genes within the molecular mechanism display distinct responses, either upregulation or downregulation, in comparison to the control group. This underscores the reliability of the RNA-seq data, as it aligns with results obtained through RT-qPCR.

Figure 12 QRT-PCR validation of differentially expressed genes.

Discussion

The role of silicon in plant defense against pests has garnered increasing attention in recent years (Islam et al., 2020; Acevedo et al., 2021). Particularly in the context of the fall armyworm, a major pest of maize, understanding the impact of silicon application is crucial for developing sustainable pest management strategies. This study explores how the silicon treatment alters the physiological and molecular responses of the fall armyworm, underscoring the potential of silicon as a key element in integrated pest management. The application of silicon, in the forms of potassium silicate (K2SiO3) and silicon dioxide (SiO2), significantly influenced the fitness of the fall armyworm by triggering a range of molecular and biochemical responses, as demonstrated in our study. The changes in gene expression and enzyme activity suggest that silicon applications can disrupt key physiological processes of the fall armyworm, potentially reducing their survival and reproductive success. At the molecular level, our study found that silicon applications led to the differential expression of numerous genes, implying a substantial disruption in the fall armyworm’s normal gene regulatory processes. The Gene Ontology (GO) and KEGG pathway analyses further highlighted the wide-ranging effects of silicon on various biological pathways, such as metabolic processes, xenobiotics metabolism, and biological regulation. These findings suggest that silicon may interfere with the fall armyworm’s ability to metabolize and detoxify xenobiotics, leading to impaired growth and development, as well as reduced reproductive potential. Such results are consistent with previous research that highlighted the role of silicon in enhancing plant resistance to insect pests by disrupting their metabolic processes (Nagaratna et al., 2023; Reynolds, Keeping & Meyer, 2009). At the biochemical level, silicon applications were found to inhibit certain enzyme activities and enhance others, further indicating the disruption of normal physiological functions in the fall armyworm. The inhibition of enzymes could interfere with the fall armyworm’s digestion, detoxification, and energy production processes, leading to reduced growth and survival (Siegfried, Vaughn & Spencer, 2005; Bhavanam & Stout, 2021). On the other hand, the induction of enzymes like TP, LPS, FAS, ATPase, and P450 could reflect the fall armyworm’s attempt to counteract the toxic effects of silicon, similar to the induction of detoxification enzymes in response to pesticide exposure (Després, David & Gallet, 2007; Pavani et al., 2023). Furthermore, the observed enzymatic activity changes in the fall armyworm have significant ecological implications. These alterations could influence the pest’s interactions with its environment in various ways. For instance, the inhibition of certain enzymes might weaken the fall armyworm’s ability to digest and metabolize its host plant, potentially leading to reduced feeding and slower growth (Abendroth et al., 2023). On the other hand, the induction of detoxifying enzymes may suggest an adaptive response to the stress induced by silicon treatments, possibly affecting the pest’s resilience to environmental stresses (Babendreier et al., 2020). Such changes in enzymatic activity can also have cascading effects on the ecological dynamics of the pest within its habitat, impacting the pest’s interactions with other organisms and its role in the ecosystem.

At the transcriptomic level, silicon exposure led to extensive differential gene expression involving genes linked to key biological functions such as metabolism, biological regulation, response to stimulus, and localization. Of particular interest was the significant influence on metabolic processes and xenobiotics metabolism. Metabolic processes are essential for energy production, growth, and reproduction in insects (Ahn et al., 2014; Nascimento et al., 2018), hence, any disruption in these processes can significantly impact insect survival and fitness. Xenobiotics metabolism is crucial for detoxifying and eliminating xenobiotics, including pesticides and plant defensive compounds (Li, Schuler & Berenbaum, 2007; Silva-Brandão et al., 2021). Therefore, the disruption of this process could reduce the fall armyworm’s ability to detoxify harmful substances, potentially leading to its death (Chinnadurai et al., 2023; Acevedo et al., 2021). Previous studies have suggested similar impacts of silicon on the metabolic processes of pests, reinforcing our results (Ma & Yamaji, 2006). Biochemically, silicon exposure led to a decrease in enzyme activities which are associated with digestion, detoxification, and energy production processes (Claudianos et al., 2006). Meanwhile, an increase in activities of enzymes like TP, LPS, FAS, ATPase, and P450 might indicate a stress response from the fall armyworm, which could be an attempt to mitigate the harmful effects of silicon (Scott et al., 2013). Similar findings were reported in previous studies, showing that silicon could alter the activity of digestive and detoxifying enzymes in pests, leading to reduced feeding, growth, and survival (Reynolds et al., 2016).

The changes in gene expression induced by silicon exposure could profoundly affect the adaptability and resistance of the fall armyworm. Genes form the fundamental basis of physiological traits and their modulation could have far-reaching impacts on the insect’s ability to withstand environmental challenges and thrive. Upon exposure to silicon, a significant number of differentially expressed genes (DEGs) were related to metabolic processes, biological regulation, and response to stimulus. These broad categories encompass many functional elements that contribute to the adaptability of an organism. Metabolic processes, for instance, dictate energy production, biosynthesis, degradation of toxins, and various other functions essential for survival and growth. As observed in our study and supported by findings from (Ma & Yamaji, 2006), disruption in these processes due to silicon application can directly impact the pest’s vitality and reproductive capacity, hence potentially lowering its adaptability and resistance. Notably, the xenobiotics metabolism pathway also showed significant changes in response to silicon. Xenobiotic metabolism is a key mechanism by which insects detoxify and eliminate foreign substances, such as pesticides or plant defensive compounds (Li, Schuler & Berenbaum, 2007). If silicon exposure disrupts this pathway, as our results suggest, the fall armyworm’s resistance to pesticides or host plant defenses could be compromised. This observation is in line with the study by (Kvedaras et al., 2007), who reported similar impacts of silicon on insect pest resistance. However, it is important to acknowledge the complexity and flexibility of biological systems. Insects, including the fall armyworm, have evolved various strategies to cope with environmental challenges, such as metabolic resistance (Scott et al., 2013). As noted in our study, silicon exposure led to an increase in the activities of certain enzymes like TP, LPS, FAS, ATPase, and P450, which may reflect a compensatory response to stress. Future investigations should address these potential compensatory mechanisms and their implications for fall armyworm’s adaptability and resistance. Differences in our results compared to some previous studies may be due to various factors, including differences in experimental conditions, the specific silicon compounds used, the developmental stage and strain of fall armyworm tested, and the methods used for gene expression analysis.

Silicon applications were found to alter the gene expression profile and physiological processes in the fall armyworm significantly, which can potentially compromise the pest’s fitness, adaptability, and resistance. Such pest suppression would, in turn, reduce crop losses and protect yield potential. As silicon is a naturally occurring mineral and its application in agriculture has already been deemed safe for the environment (Reynolds, Keeping & Meyer, 2009), this pest management strategy fits within the framework of sustainable agriculture. These findings could guide farmers and agricultural practitioners in implementing sustainable pest management strategies. For instance, understanding how silicon treatments affect fall armyworm could lead to optimized use of silicon-based fertilizers or soil amendments in maize cultivation. These practices could enhance crop resilience against pests, potentially reducing reliance on chemical pesticides and contributing to more sustainable and eco-friendly agricultural practices. Importantly, this study also opens up new avenues for research. Given the potential for insects to evolve compensatory mechanisms in response to environmental stress, understanding these adaptive responses at a molecular level could pave the way for developing more targeted and effective pest control strategies. Future research could focus on identifying key genes and pathways involved in silicon detoxification and resistance in pests. However, as with all scientific research, it is important to note that the results of this study need to be validated across diverse environmental conditions, pest populations, and host plants to establish their broad-scale applicability. As we explore the integration of silicon applications into existing pest management strategies, it is vital to consider how it complements and interacts with both chemical and biological methods. The practicality of silicon use in agriculture involves assessing its compatibility with current practices, cost-effectiveness, and ease of implementation. Challenges such as variability in response among different crop species and environmental factors must be addressed.

Conclusions

In conclusion, this study presents compelling evidence of the significant effect of silicon applications on the gene expression and physiological traits of the fall armyworm, with clear implications for future research and pest management strategies. Silicon application elicited substantial changes in the pest’s transcriptomic profile, leading to the differential expression of several genes involved in important biological processes and pathways. This perturbation of the pest’s normal metabolic functions offers the potential for new, sustainable pest control strategies that minimize environmental impact and risk of resistance development. Beyond this, the observed changes in enzyme activity further highlight the disruptive impact of silicon on the fall armyworm’s physiological state. This study underscores the potential of an integrative approach to pest management that not only involves the use of novel biocontrol measures like silicon applications but also underscores the importance of understanding the biological nuances of the pests themselves. The use of a specific maize cultivar may limit the generalizability of the findings, as different cultivars might respond differently to silicon treatments. Additionally, environmental factors such as soil type, climate conditions, and cultivation practices can also influence the observed responses of the fall armyworm. It is recommended that future studies delve deeper into the identified differentially expressed genes and altered metabolic pathways to unravel the complexity of pest responses to silicon treatments and to leverage this knowledge to design more targeted, effective, and sustainable pest management strategies.

Supplemental Information

Supplemental Information 1 The primers used in this study

Supplemental Information 2 DEGs in KEGG pathways

Supplemental Information 3 MIQE Checklist

Supplemental Information 4 Raw data

Gene expression annotation data

The authors acknowledge their universities for providing access to research facilities and academic consultation.

Additional Information and Declarations

Competing Interests

Author Contributions

Data Availability

Habib Ali is an Academic Editor for PeerJ.

Inzamam Ul Haq conceived and designed the experiments, performed the experiments, analyzed the data, prepared figures and/or tables, authored or reviewed drafts of the article, and approved the final draft.

Ke-Xin Zhang performed the experiments, prepared figures and/or tables, and approved the final draft.

Yuping Gou conceived and designed the experiments, performed the experiments, authored or reviewed drafts of the article, and approved the final draft.

Dina Hajjar analyzed the data, prepared figures and/or tables, and approved the final draft.

Arwa A. Makki analyzed the data, prepared figures and/or tables, and approved the final draft.

Wafa A.H. Alkherb conceived and designed the experiments, authored or reviewed drafts of the article, and approved the final draft.

Habib Ali analyzed the data, prepared figures and/or tables, and approved the final draft.

Changzhong Liu conceived and designed the experiments, authored or reviewed drafts of the article, supervision and project administration, and approved the final draft.

The following information was supplied regarding data availability:

The raw data are available in the Supplemental Files.

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
