# Peer review of "Transcriptomic and biochemical insights into fall armyworm (Spodoptera frugiperda) responses on silicon-treated maize"

_PeerJ, doi:10.7717/peerj.16859_

## Round 0.1 · original submission · Major Revisions

Kindly revise the manuscript in light of the comments received from all four reviewers.

Reviewer 1 ·

Basic reporting

.

Experimental design

.

Validity of the findings

.

Additional comments

Dear Editor
Thank you for providing the opportunity to review this fascinating manuscript. Overall, the study presents valuable insights regarding the impact of silicon applications on the gene expression and physiological traits of the Fall Armyworm. The research is well-structured, and the findings have the potential to contribute significantly to pest management strategies. The following constructive comments aim to enhance the manuscript's clarity, depth, and practical applicability. Each comment is intended to guide authors in refining specific aspects, with the overarching goal of further strengthening the quality and impact of the work. I look forward to seeing the manuscript progress.
1. Add specific details to the title for a clearer idea of the focus. For example, specifying the type of gene expression changes or the targeted aspects of Fall Armyworm physiology
2. Specify the primary outcomes or findings in more detail.
3. The abstract is well-structured. Consider briefly mentioning the significance of the observed changes in enzyme activities in the abstract, as this is a key finding of the study.
4. Clearly state the knowledge gap that this study aims to address concerning sustainable pest management strategies
5. The methods are well-detailed. It would be beneficial to include a sentence on the rationale for choosing the specific concentrations of silicon sources to provide context for readers
6. Provide a brief rationale for choosing the specific concentrations of Silicon Dioxide (SiO2) and Potassium Silicate (K2SiO3) used in the study
7. The Pearson’s correlation analysis is crucial. Briefly explain the biological significance of the correlation values in the context of gene expression stability within treatment groups.
8. Consider providing a brief overview or summary before diving into the details of DEGs and enzyme activity results.
9. In discussing the impact on gene expression, elaborate on how these changes might directly influence the physiological traits of the Fall Armyworm.
10. Emphasize the potential ecological implications of the observed changes in enzymatic activities. How might these alterations influence the interactions of the Fall Armyworm with its environment?
11. When discussing the GO enrichment results, briefly highlight a few specific genes or pathways within each category to illustrate the biological relevance of the findings.
12. Address the potential limitations of the study, such as the specific cultivar used or any factors that might affect the generalizability of the findings.
13. Strengthen the connection between the conclusion and the study's implications for pest management. Clearly state how the observed changes could be translated into practical, sustainable strategies.
14. Consider mentioning the next steps for future research. What specific aspects of the identified genes or pathways could be explored further to enhance our understanding?
15. Throughout the manuscript, ensure consistent use of terminology related to gene expression changes (e.g., up-regulation, down-regulation) for clarity.
16. Make sure that the rationale behind choosing Long-Fei 211 as the maize variety is explicitly stated in the Materials and Methods.
17. Clarify the potential implications of the study findings for practical applications in agriculture. How might farmers or agricultural practitioners benefit from the insights gained?
18. Consider adding a brief statement about the novelty of your study compared to previous research on silicon applications in pest management.
19. Discuss the potential environmental impacts of silicon application. Are there any considerations regarding runoff or accumulation in the soil?
20. Use consistent units throughout the manuscript, especially in the Materials & Methods and Results sections.
21. Check for clarity and coherence in the language, ensuring that complex concepts are communicated in an accessible way.
22. In the Materials and Methods, provide a brief justification for choosing the specific developmental stages of Fall Armyworm used in the study.
The authors are advised to carefully consider these suggestions for further improvement of the manuscript.

·

Basic reporting

After reviewing the assigned manuscript, I have provided some suggested changes that should be incorporated to enhance the quality of the manuscript. I strongly advise the authors to carefully consider these suggestions in their revised version.

Experimental design

My suggestions for authors are mentioned below:

Materials & Methods
Why authors use foliar sprays and avoided drenching treatments. Any particular reason for selectin Si dose (800 ppm)?
Line 95: laboratory conditions: Be more specific and mention temperature, humidity and photoperiod ranges here at the start of experimental procedure
Line 98: Also mention application method and dose rates.
Line 102: standardized pro mix potting soil. Its better to mention the ingredients with ratio. Also mention Pot size.
103: Replace degree Celsius with °C throughout the manuscript.
106: parts per million (ppm)
Avoid using terms like we isolated, we assessed
What about Statistical analysis section?
Results
The results section seems ok with author can also add exact values depending on his choice.
Discussion
Is better to add an introductory paragraph stating the importance of Si against target pest.
The few last lines of discussion section can be incorporated in conclusion section. The authors are advised to give a sound recommendation regarding Si applications in maize. What would be ideal either a single application or split applications at regular intervals? Also mention what is missing in current study which needs to be addressed in further studies.

Validity of the findings

Results
The results section seems ok and the author can also add exact values depending on his choice.
Discussion
It would be better to add an introductory paragraph stating the importance of Si against the target pest.
The few last lines of the discussion section can be incorporated in conclusion section. The authors are advised to give a sound recommendation regarding Si applications in maize. What would be ideal either a single application or split applications at regular intervals? Also mention what is missing in the current study that needs to be addressed in further studies.

Additional comments

Abstract
Biochemical analyses further revealed that Si treatments inhibited several enzyme activities while simultaneously inducing others, thus suggesting a toxic effect on the Fall Armyworm.
Its better to mention the names of enzymes or the activities involved rather than give a generic statement. Please revise the above statement for more clarity.
In conclusion of abstract, author stated Si applications
Does one application is enough or Si have to be applied at regular intervals. The authors are advised to give a sound recommendation depending on the feeding nature of the pest.
Introduction
Line 44: The Fall Armyworm, Spodoptera frugiperda
Also mention its family and order along with authority
Line 45: Replace threat in with threat to
In the text citations are not placed properly (Haq et al., 2022b; Guo et al., 2018) (Akhtar et al., 2022; He et al., 2023). Follow one sequence as per Journal guidelines i.e. from latest to old or old to latest.
Line 52: rapid lifecycle: shorter lifecycle will be more suitable
Line 64: Replace environmentally friendly with eco-friendly
The authors are advised to use more suitable and specific terms throughout the manuscript.
Line 65-66: to biotic and abiotic stress: Better to write against both biotic and abiotic stresses. Add more recent citations here separately for biotic and abiotic stresses.
Line 66: Silicon is non-toxic
Ambiguous statement as Si is toxic for insects and it can be also toxic for plants at specific higher concentrations. Be careful with your statements.
Line 66-67 and 70-71 are almost similar. Avoid these kinds of repetitions.
Line 76: Add more citations.
Line 77-85: Very well written.
Line 86: Before stating the objectives, the authors are advised to add a study gap statement. This will further enhance the significance of current study.

Reviewer 3 ·

Basic reporting

The manuscript demonstrates a commendable standard of basic reporting. The use of professional English is clear and unambiguous throughout, contributing to the overall readability of the document. The inclusion of literature references is good, but authors can add some more recent citations, providing an appropriate field background and context for the study. The article structure adheres to professional standards, ensuring a logical flow of information. Figures and tables are well-designed and effectively contribute to the presentation of results. Figure captions need to be improved. Importantly, the inclusion of raw data enhances the transparency of the study. The manuscript stands as a self-contained piece, effectively tying the presented results back to the initial hypotheses. Overall, the basic reporting aspects are robust, facilitating a comprehensive understanding of the research.

Experimental design

The experimental design of the study is well-conceived and demonstrates a strong foundation. The use of a completely randomized design (CRD) with three biological replicates enhances the reliability of the results. The detailed description of the experimental site, treatments, and plant materials provides clarity on the methodology. The controlled growth conditions for both the maize plants and Fall Armyworm larvae are adequately explained, ensuring the reproducibility of the study. The application of silicon compounds and the subsequent procedures, such as rearing and feeding of the larvae, is meticulously outlined. The inclusion of RNA extraction and sequencing, bioinformatic analysis, and validation methods ensures a comprehensive approach to data generation and interpretation. Additionally, the statistical power calculation adds to the robustness of the experimental design. Overall, the experimental design is well-structured, transparent, and aligns with the research objectives, contributing to the credibility of the study.

Validity of the findings

The validity of the findings in this study is well-supported by a systematic and rigorous approach. The utilization of Pearson's correlation analysis provides a robust assessment of the relationship between gene expression levels, offering confidence in the reliability of the data. The identification and characterization of differentially expressed genes (DEGs) through RNA sequencing, coupled with comprehensive bioinformatic analyses, contribute to the internal validity of the results. The subsequent validation of DEGs through real-time quantitative RT-PCR adds an additional layer of credibility. Moreover, the enzyme activity assays provide a valuable biochemical perspective, enhancing the overall validity of the observed effects on Fall Armyworm physiology. The consistency between the RNA-seq results and the RT-qPCR validation underscores the reliability of the gene expression data. The KEGG pathway analysis further strengthens the validity by connecting the genetic responses to broader biological processes. Overall, the study's findings are well-founded, with appropriate methodologies employed to ensure the integrity and validity of the observed outcomes.

Additional comments

I appreciate the thoroughness and scientific rigor evident in this manuscript. The study provides valuable insights into the impact of silicon applications on the gene expression and physiological traits of the Fall Armyworm. The research is well-organized, and the findings hold promise for contributing significantly to pest management strategies. Constructive comments have been provided to enhance clarity, depth, and practical applicability, aiming to further strengthen the quality and impact of the work. The commitment to advancing scientific knowledge is evident, and I anticipate that the manuscript will benefit from addressing the suggested refinements. I commend the authors for their efforts and look forward to witnessing the manuscript's progression.
My suggestion are as follows:
Include more details on the methods and results in the abstract to provide a comprehensive overview of the study.
Strengthen the transition between the general background on Fall Armyworm and the specific focus on Silicon Applications in introduction section.
Please add more citations related to the work.
Provide a concise overview of previous research on Silicon Applications in pest management to contextualize the study better.
Specify the rationale behind choosing Long-Fei 211 for the experiment and its relevance to real-world agricultural scenarios.
Clarify the criteria for selecting the specific concentrations (800 parts per million) of Silicon Dioxide and Potassium Silicate.
When discussing the KEGG Scatter Analysis, relate the observed pathways to potential impacts on the Fall Armyworm's ecological interactions.
Figure captions are very short. Please write more detailed captions.
Provide a more in-depth comparative analysis of the results with existing literature on the impact of Silicon Applications in pest management. Discuss similarities and differences to highlight the unique contributions of this study.
Expand on the potential environmental implications of silicon applications discussed in the conclusion. Consider how these findings align with broader environmental considerations and sustainable agriculture practices.
Elaborate on the concept of compensatory mechanisms briefly mentioned in the Discussion. Delve deeper into the potential adaptive responses of Fall Armyworms to silicon exposure, providing a nuanced perspective.
In the context of insect resistance, discuss the potential for metabolic resistance in Fall Armyworms against silicon applications. Address how the study's findings align with or challenge existing knowledge on insect resistance mechanisms.
Discuss how silicon applications could be integrated into existing pest control strategies, considering both chemical and biological methods. Address the practicality and challenges of incorporating silicon in real-world agricultural settings.
Explore the trade-offs in pest fitness resulting from the gene expression changes observed. Consider how alterations in metabolic pathways may impact not only survival but also reproductive success and population dynamics.
Broaden the discussion to consider potential implications for other pests beyond the Fall Armyworm. Discuss whether the observed effects might be generalizable to a broader spectrum of insect pests.

·

Basic reporting

The manuscript appears to be clear with clarity on background, data is clearly represented with good quality figures

Experimental design

EXPERIMENTAL DESIGN
Methods described with sufficient detail & information but there is a need of more clarity on number of insects used for experimental purpose.

Validity of the findings

I think the study is of first of its kind with deepens the area of induced host plant resistance against FAW. The study is robust and statistically sound.

Additional comments

MATERIAL AND METHODS ;
Line 127 ; Any reference for following this method. Quote a related reference. What is the size of leaf disc. Is it uniform for all the treatments.?
Line 132 ; Which stage larva were used? Are they of same aged/ size?
It is also not clear, how many larvae were used, replication or in groups? Needs clarity.
RESULTS ; Line 175 -177 ; The sentence should be moved to methodology.
Discussion ; Neat and clear.
REFERENCES ; Many references have been added indicating thorough revision of work carried out previously. One of the same author’s name is quoted differently in Line 521, Line 524 and Line 531. Double check.
I recommend the manuscript for acceptance with minimum revision as pointed above.

---

## Round 0.2 · accepted · Accept

Reviewers are satisfied with the revision made.

Reviewer 1 ·

Basic reporting

the authors have addressed all the comments and the paper can be accepted

Experimental design

the authors have addressed all the comments and the paper can be accepted

Validity of the findings

the authors have addressed all the comments and the paper can be accepted

Additional comments

the authors have addressed all the comments and the paper can be accepted

·

Basic reporting

No more comments from my side.

Experimental design

No more comments from my side.

Validity of the findings

No more comments from my side.

Reviewer 3 ·

Basic reporting

The manuscript 'Transcriptomic and Biochemical Insights into Fall Armyworm (Spodoptera frugiperda) Responses on Si-Treated Maize' demonstrates clear, concise, and coherent reporting, adhering to standard scientific communication guidelines effectively.

Experimental design

The experimental design of the manuscript is robust, well-structured, and methodologically sound, ensuring reliable and reproducible results in the study of Spodoptera frugiperda responses to Si-treated maize.

Validity of the findings

The findings presented in this manuscript are valid, supported by robust data and analysis, and contribute significantly to our understanding of the interaction between Spodoptera frugiperda and Si-treated maize.